# Mental Health Issues among Caregivers of Young Children in Rural China: Prevalence, Risk Factors, and Links to Child Developmental Outcomes

**DOI:** 10.3390/ijerph18010197

**Published:** 2020-12-29

**Authors:** Siqi Zhang, Lei Wang, Yue Xian, Yu Bai

**Affiliations:** 1Center for Experimental Economics of Education, Shaanxi Normal University, Xi’an 710119, China; zhangsiqiceee@163.com; 2International Business School, Shaanxi Normal University, Xi’an 710119, China; xianyueceee@163.com; 3School of Economics, Minzu University of China, Beijing 100081, China; yubai@muc.edu.cn; 4China Institute for Vitalizing Border Areas and Enriching the People, Beijing 100081, China

**Keywords:** caregiver’s mental health, risk factors, child development, rural China

## Abstract

Previous research has found that there are high rates of mental health problems among caregivers in rural China and that caregiver mental health may be a significant predictor of developmental delays among infants and toddlers in these rural areas. In this paper, we use data from a survey of 986 caregiver-child pairs in rural China to examine the risk factors of caregiver mental health and measure the association between caregiver mental health and child development outcomes. To conduct the empirical analysis, we assess caregiver mental health using the Depression Anxiety Stress Scales-21 (DASS-21) questionnaire and measure child developmental outcomes using the Bayley Scales of Infant and Toddler Developmental Third Edition (Bayley-III). The results show that 32% of caregivers have depressive symptoms, 42% have anxiety symptoms, and 30% have symptoms of stress. The data also demonstrate that caregiver identity and age as well as different measures of socioeconomic status (SES) (characterized by caregiver education, father’s education, and household wealth) are all significantly linked to symptoms of caregiver mental health problems. The analysis shows that caregiver depression, anxiety, and/or stress are significantly associated with lower early childhood development (ECD) outcomes in all areas measured (cognition, language, social-emotion, and motor skills). The heterogeneous analysis demonstrates that there are differences in the effects of caregiver mental health problems on ECD among households are from families that are endowed with different levels of SES. On the basis of the findings the study concludes that policymakers should pay more attention to caregiver mental health problems in order to improve child developmental outcomes in rural China. The study cannot, however, draw casual conclusions and cannot rule out the possibility of recall bias when measuring caregiver mental health, which may limit the external validity of the findings.

## 1. Introduction

Early childhood development (ECD) has a lifelong impact on the skill development of individuals, and it is an important cornerstone of sustainable social and economic development [1,2,3]. The first three years of life are recognized as a critical period of neurodevelopment and synaptic formation, and cognitive and non-cognitive development during this period of time provides the foundation for long-term skills [2,4,5,6]. The literature has shown that there is a significant association between better developmental outcomes in early childhood (i.e., stronger cognitive and non-cognitive skills) and higher levels of educational attainment and income later in life [5,7,8,9]. In contrast, worse developmental outcomes in early childhood (i.e., weaker cognitive and non-cognitive skills) are associated with higher rates of adult unemployment and criminal activity [9,10,11]. Researchers also have found that on a national level, investments in ECD produce greater long-run returns to human capital and economic growth than do investments in later skill development [2,6].

ECD is influenced by many factors, but a key input is the mental health of the primary caregiver, often the child’s mother. Caregivers, usually mothers, have the closest and most frequent contact with young children during their critical period of growth, and play a critical role in child health and ECD [11,12,13,14,15,16]. Multiple studies have found significant associations between maternal depressive symptoms and impaired child health, growth and development [17,18,19,20]. These associations have been attributed to lower quality caregiving among mothers with mental health issues: A meta-analysis of 46 observational studies found that depressed mothers tend to be more disengaged, irritable or hostile toward their child as well as less likely to engage their children in stimulating activities [21].

Although caregiver mental health issues are relatively widespread across the world, there is evidence that caregivers may be more vulnerable to mental health issues in low- and middle-income countries (LMICs). According to the World Health Organization (WHO), globally, about 10% of pregnant woman and 13% of woman who have just given birth have experienced a mental disorder, primarily depression. In developing countries, however, this rate is much higher, at 15.6% during pregnancy and 19.8% after childbirth [22]. A review paper focused on postnatal depression found that of the 28 LMICs where studies had been conducted, 22 countries had average prevalence estimates that were higher than high-income settings [15].

The international literature also has found multiple risk factors associated with mental health issues among caregivers. Research by the WHO and others has shown that in both developed and developing countries, caregiver mental health is closely related to economic adversity and poverty-related indicators, including low educational levels, low incomes, and poor housing conditions [15,23]. Research has similarly found that, in LMICs, lower educational level and poor family financial status are risk factors for caregiver mental health issues [24,25,26]. Other risk factors for caregiver mental health problems in LMICs are older age, poor health, lack of social support, raising multiple children and raising girls [20,26].

In rural areas of China, recent studies have begun to examine the prevalence of mental health issues among caregivers and their effects on the ECD outcomes of young children. According to recent studies, the prevalence of mental health issues among caregivers is higher in rural areas of China compared to many other developing countries [27,28]. Previous studies demonstrated that mental health problems affect between 23% and 40% of caregivers of young children in rural China [29,30,31,32]. This is higher than rates of caregiver mental health issues in other LMICs such as India (23%) [33] or Vietnam (21%) [34]. 

The existing literature has identified, as well, several risk factors for mental health issues among mothers and caregivers in rural China. Qualitative evidence points to poverty-related factors as potential risk factors for depression among rural caregivers [30,35]. In addition, factors such as poor physical health [36], raising multiple children [37], and a husband’s not living at home [38] have been found to be potential risk factors for mental health issues among rural mothers in China. 

Recent studies in rural China also have found that maternal mental health is associated with developmental delays among young children. Two studies have found that caregiver depressive symptoms are linked with significantly lower developmental scores among young children in rural communities [30,39]. An additional study of depression, anxiety and stress among caregivers across rural China found significant associations between mental health issues and lower ECD outcomes [31].

Although the literature both in worldwide and in China has provided a large body of evidence on the mental health issues of mothers, gaps in the literature on caregiver mental health still remain. In particular, less is known about the mental health issues of non-mother caregivers, such as grandmothers. In rural China, a large share of caregivers of infants and toddlers is grandmothers, who have taken over caregiving duties due to parental out-migration [40]. Although there is some evidence that grandmothers may be at higher risk for mental health issues compared to mothers [30], however the literature to date is limited and inconclusive. Further, although previous studies have shown that mental health may interact with other risk factors (such as SES) to affect the ECD outcomes of young children [21,41,42,43], little is known about the interactions of mental health and SES on ECD in rural China.

This study seeks to address the gaps in the literature by empirically examining the prevalence, correlates, and consequences of mental health issues among caregivers of infants and toddlers in rural China. In order to achieve this goal, we have four specific objectives. First, we describe the prevalence of caregiver mental health problems in rural China, focusing on depression, anxiety and stress. Second, we measure the risk factors that are associated with symptoms of each mental health issue. Third, we describe the ECD outcomes of young children in rural China and examine correlations between caregiver mental health problems and the ECD outcomes of their children. Fourth, we identify the heterogeneous effects of caregiver mental health problems on ECD outcomes by SES.

## 2. Materials and Methods

### 2.1. Ethical Approval

This study received ethical approval from the Stanford University Institutional Review Board (IRB) (Protocol ID 50901), and from the Kunming Medical University Ethical Review Board. All participating caregivers gave their oral consent for both their own and their infant’s involvement in the study.

### 2.2. Study Location and Sampling

The data presented in this study were collected in a nationally-designated rural poverty county in a southwestern province of China. The sample province is a remote mountainous region prone to natural disasters [44]. The province is also one of China’s poorest regions. In 2019, the per capita GDP of the province was $7067 (RMB 47,944), far lower than the national per capital GDP of $10,394 (RMB 70,892). It ranked 24th out of 31 provinces in mainland China in terms of provincial GDP per capita [45].

The sampling strategy for our survey was as follows. First, we randomly chose two townships within the sample county. We then used official government data to compile a list of villages from each township and included all villages in our sampling frame. Finally, with the assistance of the local family planning official in each township, a list of all registered births over the past 24 months was obtained in each village. All children in our target age range (6–24 months) and their caregivers were enrolled in the study. Overall, the study included 986 caregiver-child pairs in 189 villages. Because the study was conducted in one province, the results may not be generalizable to other regions. In addition, due to its cross-sectional design, our study does not allow us to identify causal relationships.

### 2.3. Data Collection

Data were collected by trained survey enumerators in March 2019. The survey collected data on mental health of caregivers as well as child ECD outcomes. We also collected data on the demographic characteristics of all sample children and households. 

#### 2.3.1. Caregiver Mental Health

We administered the Depression Anxiety Stress Scale 21 (DASS-21) to all primary caregivers. The DASS-21 is a short version of the 42-item self-report measure of DASS. The scale includes 21-items, grouped into three subscales that measure symptoms of depression, anxiety and stress. Although the DASS-21 cannot be interpreted as a tool for direct clinical diagnosis [46], it is designed to be a quantitative measure of the severity of depression, anxiety, and the stress symptoms. The DASS-21 has proven construct validity and high reliabilities [47] and relies on self-reporting of symptoms by caregivers, which cannot rule out the possibility of recall bias. The Chinese version of the DASS-21 was translated into Chinese by Zuo and Chang (2008) [48]. In China, researchers established the cross-cultural validity of the DASS-42 and the validity for the DASS-21 [49,50]. 

Primary caregivers were asked to respond to each item of the DASS-21 by rating the frequency and severity of experiencing symptoms over the previous week using a 4-point Likert scale (0 = It did not apply to me at all; 1 = It applied to me to some degree, or some of the time; 2 = It applied to me to a considerable degree, or a good part of time; 3 = It applied to me very much, or most of the time). For each subscale of the DASS-21, items were scored 0, 1, 2 or 3 respectively. Scaled scores for the DASS-21 subscales of depression, anxiety, and stress were derived by totaling the scores for each subscale and multiplying by two. Using cutoffs established in previous studies [49,50], we divided participants into the following categories based on the severity of symptoms: normal (0–9 for depression, 0–7 for anxiety, and 0–14 for stress), mild (10–13 for depression, 8–9 for anxiety, and 15–18 for stress), moderate (14–20 for depression, 10–14 for anxiety, and 19–25 for stress), severe (21–27 for depression, 15–19 for anxiety, 26–33 for stress), and extremely severe (≥28 for depression, ≥20 for anxiety, and ≥34 for stress). In this study, we consider caregivers to have symptoms of a mental health issue if they scored at or above the cutoff for mild symptoms (10 for depression, 8 for anxiety, and 15 for stress). 

#### 2.3.2. Early Childhood Development

All children were administered the Third Edition of the Bayley Scales of Infant and Toddler Development (Bayley-III), an internationally recognized method of assessing ECD [51]. The Bayley-III is generally considered to be the gold standard for assessing ECD outcomes among children ages 1–42 months. The results are categorized into five standardized scales, four of which we sued in the present study: cognitive, language, social-emotional, and motor. Studies that examine the validity of the Bayley-III have found that the four scales exhibit high inter- and intra-rater reliability agreement, high internal consistency, and high test-retest stability in multiple cultural contexts [52,53,54,55]. Bayley-III was translated and adapted for Chinese settings by Xu et al., (2011) [56]. All enumerators underwent a formal weeklong training course on how to administer the test, including a 2.5-day experiential learning component in the field. Enumerators administered the test in the hone of each child, using a set of standardized toys and a detailed scoring sheet.

The cognitive, language, and motor scales were administered by trained enumerators who evaluated the child based on his or her performance on a number of tasks. The social-emotional scale was administered by asking the child’s primary caregiver a series of questions to assess the child’s mastery of functional emotional skills, such as self-regulation and interest in the world; communicating one’s needs; engaging with others and establishing relationships; using emotions in an interactive, purposeful manner; and using emotional signals or gestures to solve problems [57]. Each of the four subscales considers the child’s gestational and chronological ages in calculating final scores. Following Bayley-III guidelines [57], raw scores were scaled by age, and then converted into standardized scores with a given mean and standard deviation according to Bayley-III guidelines. 

We also examined the prevalence of developmental delays for the entire sample. We define delays according to documented distributions of Bayley-III scores in Bayley manual and reference populations. The mean score (SD) is expected to be 105 (9.6) for the cognitive scale [58,59], 109 (12.3) for the language scale [59], 100 (15) for the social-emotional scale [57], and 107 (14) for the motor scale [58,60]. Children with Bayley-III subscale scores more than −1SD below the reference mean are considered delayed.

#### 2.3.3. Demographic Characteristics

We collected child and household characteristics for each family. Child characteristics included age (in months), gender, ethnicity, whether the child was born prematurely, and whether the child has siblings. The age of the child was taken from his/her birth certificate. Household characteristics included the identity of the primary caregiver (e.g., mother, grandmother, other), the age and educational level of the primary caregiver, the educational level of the child’s father, and whether the father lived at home with the child (or lived out of the village as a migrant worker). The survey team also collected data to measure the value of household assets. A household asset index was constructed using polychoric principal components for whether the household owned the following items: a flush toilet, water heater, computer, internet, air conditioner, and car.

### 2.4. Statistical Analysis

Our statistical analysis includes four parts. We first examine the prevalence of depression, anxiety, and stress among the full sample, as well as among subgroups of caregivers based on child and household demographic characteristics (child’s age, gender, ethnicity, and premature birth; whether the child has siblings; whether the mother is the primary caregiver; caregiver age and educational level; father’s educational level; whether the father is at home; and household asset index). To understand which subgroups, have a higher incidence of mental health problems, we also compared the shares of caregivers with depression, anxiety, and stress symptoms across subgroups. 

Second, in order to identify the risk factors for mental health issues among caregivers, we constructed a model as follows: (1)$${\mathrm{Mental}\;\mathrm{Health}}_{i}=\beta_{0}+\beta_{1}{\mathrm{risk}\;\mathrm{factor}}_{i}+\varepsilon_{i}$$
where the dependent variable, ${\mathrm{Mental}\;\mathrm{Health}}_{i}$, represents the DASS-21 subscale score (depression, anxiety, or stress) of the caregiver of infant i. The variable ${\mathrm{risk}\;\mathrm{factor}}_{i}$ represents child and household demographic characteristics, including child’s age, gender, ethnicity, and premature birth; whether the child has siblings; whether the mother is the primary caregiver; caregiver age and educational level; father’s educational level; whether the father lives at home; and household asset index. 

Third, we measure the correlation between caregiver’s mental health and ECD outcomes using the following model:(2)$${\mathrm{Developmental}\;\mathrm{Outcomes}}_{i}=\beta_{0}+\beta_{1}{\mathrm{mental}\;\mathrm{health}}_{i}+\beta_{2}X_{i}+\varepsilon_{i}$$
where the dependent variable, ${\mathrm{Developmental}\;\mathrm{Outcomes}}_{i}$, indicates the Bayley-III subscale score (cognitive, language, social-emotional or motor scores) of infant i. ${\mathrm{Mental}\;\mathrm{health}}_{i}$ is a dummy variable is equal to 1 if the caregiver is symptomatic of a mental health problem (depression, anxiety, stress, or any of the three) and 0 if otherwise. The term $X_{i}\;$ is a vector of covariates that capture the demographic characteristics, including child’s age, gender, ethnicity, and premature birth; whether the child has siblings; whether the mother is the primary caregiver; caregiver age and educational level; father’s educational level; whether the father lives at home; and household asset index. The model also controls for Bayley-III tester fixed effects.

Finally, we conduct a heterogeneous analysis to measure the effects of caregiver mental health problems on ECD outcomes by three SES characteristics (i.e., by the educational level of the caregiver; by the educational level of the child’s father or paternal educational; and by household wealth measured by assets). To perform the heterogeneous analysis, we constructed a model as follows:(3)$${\mathrm{Developmental}\;\mathrm{Outcomes}}_{i}=\beta_{0}+\beta_{1}\varphi_{i}+\beta_{2}\delta_{i}+\beta_{3}\eta_{i}+\beta_{4}X_{i}+\varepsilon_{i}$$
where $\varphi_{i}\;,{\;\delta }_{i},\;and{\;\eta }_{i}\;$ are three dummy variables. $\varphi_{i}\;$ equals to 1 when the caregiver has symptoms of mental health problems and low SES (i.e., low caregiver educational level/low paternal educational level/low household wealth), and 0 otherwise; $\delta_{i}$ equals to 1 when the caregiver has symptoms of mental health problems and high SES (i.e., high caregiver educational level/high paternal educational level/high household wealth), and 0 otherwise; $\eta_{i}$ equals to 1 when the caregiver does not have symptoms of mental health problems and low SES, and 0 otherwise. The term $X_{i}\;$ is a vector of covariates that capture the demographic characteristics as the same as model (2).

Caregiver mental health measures, ECD outcomes and demographic characteristics were analyzed using mean and standard deviation. All correlational analyses were performed using STATA 15.1. (StataCorp LLC, College Station, TX, USA). Standard errors account for clustering at the village level, and *p*-values below 0.05 were considered statistically significant.

## 3. Results

This section presents our descriptive and correlational analyses. First, we present the descriptive statistics for the study’s children and households. Second, we describe the mental health outcomes (depression, anxiety, and stress) among the sample caregivers and the ECD outcomes of sample children. Third, we measure the risk factors that are associated with caregiver symptoms of mental health problems in rural China. Fourth, we examine the associations between caregiver mental health and ECD outcomes. Finally, we identify heterogeneous associations between caregiver symptoms of mental health problems and ECD outcomes by SES (i.e., primary caregiver educational level, father’s educational level, and household assets). The purpose of the final analysis is to determine whether the relationship between caregiver mental health and ECD outcomes differs when children and families have different levels of SES.

### 3.1. Demographic and Socioeconomic Characteristics

Table 1 shows the basic demographic and socioeconomic characteristics of study participants. Among the children in our sample, around half (52%) were male, and 69% had siblings. Only a small proportion of the children (5%) were premature. The data show that 91% of the sample children were of Han nationality, and the average age was 16 months. With regard to the household characteristics of the sample respondents, for 79% of the sample children, the mother was the primary caregiver. In the case of the remaining 21% of the caregivers, the paternal grandmother was most often the primary caregiver. Among primary caregivers in our sample, the average age was 32 years old. Over half of primary caregivers (58%) had completed less than 9 years of schooling. In the case of 28% of the children, the father was at home and did not out-migrate for work. Among fathers, over half (55%) had not attained 9 years of schooling.

### 3.2. Caregiver Mental Health Outcomes

Overall, a large share of primary caregivers in our sample had symptoms of depression, anxiety, or stress (Table 2). Specifically, 32% of primary caregivers scored above the threshold for depressive symptoms, 42% scored above the threshold for anxiety symptoms, and 30% scored above the threshold for stress symptoms. More than half of sample primary caregivers (53%) had symptoms of at least one mental health problem (i.e., depression, anxiety, or stress). When considering the severity of caregiver mental health problems, we found that for all caregivers, 11% had mild depressive symptoms; 13% showed moderate symptoms, 5% showed severe depressive symptoms, and 4% showed symptoms of extremely severe depression. When dividing the full sample by the severity of anxiety symptoms, we found that 11% had mild symptoms, 15% showed moderate symptoms, 6% had severe symptoms, and 10% showed extremely severe symptoms. When dividing the full sample by the severity of stress symptoms, we show that 10% of caregivers had mild stress symptoms, 11% had moderate symptoms, 7% had severe symptoms and 3% had extremely severe symptoms. Finally, when considering caregivers of at least one mental health problem (i.e., depression, anxiety, or stress), we found that 27% had mild symptoms, 30% had moderate symptoms, 14% had severe symptoms, and 11% had extremely severe symptoms.

### 3.3. Risk Factors of Caregiver Mental Health Problems

Table 3 and Table 4 present two ways to examine the risk factors that are associated with caregiver mental health problems. Table 3 provides a comparison of the prevalence of caregiver mental health problems (depression, anxiety, and stress) between different subgroups. The results show no statistically significant differences in the prevalence of caregiver mental health problems between subgroups divided by child gender, age, and ethnicity; whether the child was premature; and whether the child had siblings. When comparing sample caregivers by household characteristics, however, the results show that, when the primary caregiver is the paternal grandmother (and not the mother), when the primary caregiver is older, when the primary caregiver is less educated, when the father of the child has lower levels of education, and when the household is poorer, the caregiver is more likely to be at risk of mental health problems.

Table 4 provides the results of an analysis of the risk factors for caregiver mental health problems by regressing child and household characteristics on caregiver mental health outcomes (i.e., depression, anxiety, and stress scores), using the multivariate model specified in Equation (1). The results of the regression provide the following findings: Although child characteristics (age, gender, ethnicity, prematurity, and whether the child has siblings) were not significantly associated with caregiver depression, anxiety, or stress scores), household characteristics are significantly correlated with caregiver mental health scores. Specifically, caregivers who are not the mother of the child (i.e., paternal grandmothers) showed significantly higher levels of anxiety (*p* < 0.05; Column 2, Row 6) and stress symptoms (*p* < 0.05; Column 3, Row 6). Fathers’ having less than 9 years of education was associated with higher levels of depressive symptoms (*p* < 0.01 Column 1, Row 9), caregivers from poorer household showed higher levels of depressive symptoms (*p* < 0.01; Column 1, Row 11) and anxiety symptoms (*p* < 0.01; Column 2, Row 11). We found no significant relationships between other household characteristics and caregiver mental health outcomes.

### 3.4. Caregiver Mental Health and Early Childhood Development

Table 5 presents the ECD outcomes of the 986 young children in the sample. The results show overall high rates of developmental delays among sample children. Specifically, 52% of children exhibit cognitive delay, 53% have language delay, and 51% have social-emotional delay. The average rate of motor delay is 31%. Among all the children in our sample, 86% exhibit at least one kind of developmental delay, and 60% of the overall sample exhibit at least two kinds of developmental delays. Approximately one-third of young children in our sample (33%) exhibit at least three kinds of developmental delays, and 9% suffer from developmental delays in all four areas measured.

The relationship between caregiver mental health problems and child development outcomes is shown in Table 6. The results indicate that children of caregivers who have symptoms of mental health problems (depression, anxiety, or stress) show significantly lower ECD outcomes. Symptoms of depression (Row 1) were associated with significantly lower scores for cognition, language, social-emotion, and motor development (all *p* < 0.01). Anxiety symptoms (Row 2) were linked to lower motor scores (*p* < 0.05), and stress symptoms were associated with significantly lower scores in child language development (*p* < 0.01) and motor development (*p* < 0.05). Finally, caregivers’ having symptoms of at least one mental health problem (depression, anxiety, or stress) was associated with significantly lower developmental scores on all scales: cognition (*p* < 0.05), language (*p* < 0.05), social-emotional (*p* < 0.05), and motor scores (*p* < 0.01) (Row 4).

### 3.5. Heterogeneous Analysis

Table 7 presents a heterogenous analysis of caregiver mental health problems on ECD outcomes by three SES characteristics: the educational level of the caregiver, the educational level of the child’s father, and household asset index. For each SES characteristic (caregiver educational level in Panel A; father’s educational level in Panel B; and household wealth in Panel C), we divide the sample into four groups: (a) caregivers with symptoms of any mental health problem and low SES (i.e., low caregiver education, low father’s education, or low household wealth); (b) caregivers with symptoms of any mental health problem and high SES (i.e., high caregiver education, high father’s education, or high household wealth); (c) caregivers without symptoms of mental health problems and low SES (i.e., low caregiver education, low father’s educational level, low household wealth); and (d) caregivers with no symptoms of mental health problems and high SES (i.e., high caregiver education, high father’s educational level, or high household wealth). Using the final dummy variable as the reference group against which the other groups are compared, we then regress developmental scores for each ECD outcome (cognition, language, social-emotion, and motor) against three dummy variables. Using this approach, we are able to test whether there are differences in the nature of the relationship between caregiver mental health and ECD outcomes when households have high or low levels of the three SES variables. 

The two specific tests that we are most interested in are: (a) whether caregivers with symptoms of any mental health problems and low levels of SES have stronger associations with lower development scores than do caregivers with symptoms of any mental health problems who have higher levels of SES (which is tested by comparing coefficients in Row 1 versus Row2; row 6 versus Row 7; Row 11 versus Row 12; see *p*-values for these tests in Rows 4, 9, and 14); and (b) whether caregivers with symptoms of any mental health problems and low levels of SES have stronger associations with lower development scores than do caregivers with no symptoms of any mental health problems but also with lower levels of SES (which is tested by comparing coefficients in Row 1 versus Row 3; Row 6 versus Row 8; Row 11 versus Row 13; see *p*-values for these tests in Rows 5, 10, and 15).

As seen in Table 7, the heterogeneous analysis for the case of the cognitive scores of children in rural families can be seen by looking at the results in Column 1 for Panels A, B, and C. The results show that there are statistically significant differences in one case of our tests of interest. Specifically, among primary caregivers with low levels of education, caregivers with symptoms of mental health problems overall had children with significantly lower cognitive scores than did caregivers with no symptoms of mental health problems (Row 5, *p* = 0.039). When comparing the relationship between the caregiver with symptoms of mental health problems and child cognitive skills among caregivers with low versus high levels of education of children, however, there is no statistically significant difference between the two groups (Panel A, Row 4, *p* = 0.070). In the case of the heterogeneous effects between the relationship of a caregiver with symptoms of mental health problems and the cognitive scores of their children by the educational levels of their fathers, there are also no significant effects (Panel B, Row 9, *p* = 0.266; Row 10, *p* = 0.162). Likewise, there are no significant difference in relation of caregiver with symptoms of mental health problems to child cognitive scores by household asset level (Panel C, Row 14, *p* = 0.189; Row 15, *p* = 0.072).

The results of the heterogeneous analysis of child language scores are presented in Row 2. When comparing the language scores of children of caregivers with symptoms of mental health problems and with low and high levels of education, the results indicate that children of caregivers with symptoms of mental health problems and low levels of education have significantly lower language scores than children of caregivers with symptoms of mental health problems and high levels of education (Panel A, Row 4, *p* = 0.018). The results also show that children of caregivers with symptoms of mental health problems and low levels of education have significantly lower language scores than do children of caregivers without symptoms of mental health problems who also have low levels of education (Panel A, Row 5, *p* = 0.035). Similar to the results for cognitive skills, there were no differences between groups related to father’s education (Panel B, Row 9, *p* = 0.323; Row 10, *p* = 0.078) or household wealth (Panel C, Row 14, *p* = 0.160; Row 15, *p* = 0.163). 

The heterogeneous analysis of caregiver/father’s education and household wealth on the social-emotional skills and motor skills of children whose caregivers have symptoms of mental health problems are presented in Columns 3 and 4, respectively. In contrast to child cognitive and language skills, the results show no significant differences in the relationship between caregiver with symptoms of mental health problems and child social-emotional or motor skills by high/low caregiver education (Panel A, Columns 3 and 4, Row 4, *p* = 0.379/0.154). Although the results show significant differences in child motor skills between caregivers with and without symptoms of mental health problems and low caregiver education (Panel A, Column 4, Row 5, *p* < 0.01), there are no significant differences in social-emotional skills (Panel A, Column 3, Row 5, *p* = 0.437). Similarly, the results find insignificant differences of caregiver with symptoms of mental health problems by high/low father’s education (Panel B, Columns 3 and 4, Row 9, *p* = 0.998/0.557) and by high/low household wealth (Panel C, Columns 3 and 4, Row 14; *p* = 0.070/0.891). There are, however, significant differences in child social-emotional and motor skills between caregivers with symptoms of mental health problems and without such symptoms and low father’s education (Panel B, Columns 3 and 4, Row 10, *p* = 0.027/<0.01), and there are differences between caregivers with and without symptoms of mental health problems and low household assets (Panel C, Columns 3 and 4, Row 15, *p* = 0.034/0.031). 

## 4. Discussion

This study examines the prevalence and risk factors of mental health problems, as well as consequences for ECD, among caregivers of young children in rural China. Using data from 986 children aged 6–24 months and their primary caregivers in southern rural China, we describe the prevalence and severity of caregiver depression, anxiety, and stress symptoms as well as examine the risk factors associated with each mental health problem. We also examine the correlations between each mental health problem and the ECD outcomes and the heterogeneous effects of caregiver mental health problems on ECD outcomes by measures of SES. 

Overall, the prevalence of caregiver mental health problems in our study (i.e., 32% for depression, 42% for anxiety, and 30% for stress) is much higher than the average rate of the worldwide population (4.4%) [61] and other developing settings (15.6–19.8%) [22]. In our sample the share of caregivers with symptoms of depression, anxiety, and stress are 32%, 42%, and 30%, respectively. The findings are also somewhat higher than previous studies conducted in other rural areas of China [30,32,36], which have found that the rate of depression is around 23%. Clearly, if our findings of having symptoms of mental health problem are illustrative of the levels of caregivers that actually have mental health issues, this indicates that caregiver mental health is a serious issue in rural areas of China.

When examining which risk factors are associated with caregiver mental health problems, the analysis finds that caregiver identity, age, and SES are all significantly linked to symptoms mental health problems. First, the results demonstrate that caregivers who are not the mother (i.e., who are the paternal grandmother) and older caregivers are more likely to exhibit symptoms of at least one of the mental health problems measured (depression, anxiety, or stress). Second, indicators of low SES, including low caregiver education, low father’s education, and low family wealth, also are associated with an increased likelihood of having at least one mental health problem. In both cases, these results are largely consistent with the findings of other studies in rural China [32,36] and other developing countries [62,63,64]. In particular, low SES has been identified as a common risk factor for mental health problems in rural China and LMICs more broadly. For example, Gan et al. and Yang et al. found that, in rural China less–educated caregivers and caregivers from poorer families are more likely to suffer from depression [32,36]. In addition, meta–analysis research of 115 studies of mental health in LMICs found that, when caregivers have lower levels of education or face poverty constraints, there is a higher probability of having depression [64]. Our findings, however, are inconsistent with some other studies that have indicated that SES is not correlated with maternal depression [65,66]. For example, a study in the U.S. found that there is no significant association between mother’s education and maternal depression [65].

The results also find that caregiver depression, anxiety, or stress are significantly associated with lower ECD outcomes in all areas measured (cognition, language, social–emotion, and motor skills). Such a relationship between caregiver mental health problems and child development outcomes is consistent with previous studies in rural China [29,30,31]. For example, a study conducted in a northwestern province of rural China found that caregiver depression is significantly associated with decreased child language and social–emotional development [30]. Our findings also are consistent with the literature from other developing countries, which find caregiver mental health problems to be associated with lower–quality caregiving and a reduced parental stimulation [15,63,67,68,69]. In South Africa, for example, depressed mothers are shown to be less involved, less sensitive, and more negative when rearing their young children, which led to worse child development outcomes [68]. Similarly, in a study of 221 infants and their caregivers in Bangladesh, the research team found that infants whose mothers had depressive symptoms developed had worse cognitive, motor, and orientation/engagement skills compared to their peers whose mothers were not suffering from depression [67]. The results of our study, however, are not in line with findings in some other studies, either in China or internationally [30,31,70]. A 2018 study conducted in rural China found that there is no significant link between caregiver depression and child cognitive development [30]. In rural Ethiopia, a study found that maternal mental health is associated only with certain domains of child development outcomes. This study found that children of mothers with symptoms of anxiety scored lower in social emotional development but did not show lower motor or language skills than children of mothers without anxiety [70].

Our heterogeneous analysis shows that there are differences in the effects of caregiver mental health problems on ECD when the households are from families who have different levels of SES. Our results indicate that children of caregivers who have symptoms of mental health problems and low SES (characterized by low caregiver education, low father’s education, or low household wealth), score lower in cognitive, language, social–emotional, and motor scales of development outcomes when compared to children of caregivers who have mental symptoms of health problems but who have higher levels of SES. Although this is the first study to estimate these heterogeneous effects in rural China, the findings are consistent with previous studies in the international literature [21,41,42,43], which has found that poverty and low levels of education among caregivers and their families tend to moderate the effects of maternal depression on ECD outcomes. In other words, children are more (less) negatively affected by caregiver mental health issues when their caregivers have lower (higher) levels of education or household wealth.

The results of this study have several implications for both policymakers and researchers in rural China. Considering the high rates of caregiver symptoms of mental health problems and child developmental delays in our sample and other rural areas of China, policymakers must increase their efforts to improve the mental health of caregivers, which in turn will lead to improved ECD outcomes. Our findings show that when the mother is not the primary caregiver and when the caregiver (mother or grandmother) lives in a poorer family, the caregiver has a higher propensity to develop symptoms of mental health problems. Further, low SES contributes to the negative effects of caregiver mental health on ECD outcomes. Based on these findings, policymakers should develop incentives that will help mothers to be able to stay at home and take a primary role in the care of the child. Financial support for mothers (or families with young children) also might be a way to reduce the rate of caregiver symptoms of mental health problems as well as reduce the consequences of such symptoms of mental health problems for ECD outcomes. In addition, given the role of education in moderating the negative effects of poor mental health on child development outcomes, educational campaigns, such as broad–based parental training programs [71,72], may improve ECD outcomes among children of caregivers with and without symptoms of mental health problems. The results would support efforts of policymakers to expand support for broad–based parenting intervention initiatives.

## 5. Conclusions

This study makes two key contributions to the literature. First, this is the first study conducted in rural China to analyze the risk factors associated with a variety of caregiver symptoms of mental health problems, including not only depression but also anxiety and stress. Second, this study also is the first to use data collected in rural China to show the heterogeneous effects of caregiver symptoms of mental health problems and SES on the developmental outcomes of young children. Our finding that SES characteristics moderate the negative effects of mental health problems on ECD outcomes can aid researchers and policymakers in their efforts to design more effective and cost–effective programs to caregiver mental health problems and improving ECD outcomes in rural China.

We also acknowledge three limitations of this study. First, this study relies on a cross–sectional data, which do not allow us to draw causal conclusions on the relationship between caregiver mental health problems and child development outcomes. Second, the data we collected on caregiver mental health (i.e., DASS–21) rely on self–reporting by caregivers; thus, we cannot rule out the possibility of recall or self–reporting biases. Third, even though the findings of this study were consistent with previous research, the study was conducted in only one province in China. It is possible that the sample is not representative of all of rural China, which limits the external validity of the findings. Future research should examine rural households across rural China to help provide a more complete understanding of the state of caregiver mental health across rural China.

## Figures and Tables

**Table 1 ijerph-18-00197-t001:** Summary statistics, *n* = 986.

	Frequency (*n*)	Percentage (%)/Mean ± SD
	(1)	(2)
***Child characteristics***		
Age (months)	986	15.65 ± 5.60
Gender		
Male	513	52
Female	473	48
Premature		
Yes	48	5
No	938	95
Han nationality		
Yes	894	91
No	92	9
Have siblings		
Yes	681	69
No	305	31
***Household characteristics***		
Primary caregiver		
Mother	777	79
Others	209	21
Caregiver age (years)	986	31.99 ± 12.08
Caregiver educational level		
under 9 years	574	58
above 9 years	412	42
Paternal educational level		
under 9 years	538	55
9 years and above	448	45
Father at home		
Yes	280	28
No	706	72
Household asset index	986	−0.00 ± 1.10

Note. Data source is author’s survey.

**Table 2 ijerph-18-00197-t002:** Mental health severity in caregivers, *n* = 986.

		Frequency (*n*)	Percentage (%)
		(1)	(2)
***Depression***			
**(1)**	**Total**	**319**	**32**
(2)	Mild	110	11
(3)	Moderate	127	13
(4)	Severe	47	5
(5)	Extremely severe	35	4
***Anxiety***			
**(6)**	**Total**	**413**	**42**
(7)	Mild	108	11
(8)	Moderate	147	15
(9)	Severe	60	6
(10)	Extremely severe	98	10
***Stress***			
**(11)**	**Total**	**296**	**30**
(12)	Mild	94	10
(13)	Moderate	107	11
(14)	Severe	68	7
(15)	Extremely severe	27	3
***Depression or Anxiety or Stress***			
**(16)**	**Total**	**522**	**53**
(17)	Mild	264	27
(18)	Moderate	292	30
(19)	Severe	138	14
(20)	Extremely severe	111	11

Note. Data source is author’s survey.

**Table 3 ijerph-18-00197-t003:** Distribution of caregiver’s mental health severity by demographic characteristics, *n* = 986.

		Depression (1 = Yes)		Anxiety (1 = Yes)		Stress (1 = Yes)	
	Observations	Frequency (Percentage)	*p*-Value	Frequency (Percentage)	*p*-Value	Frequency (Percentage)	*p*-Value
	(1)	(2)	(3)	(4)	(5)	(6)	(7)
***Child characteristics***							
Age (months)			0.575		0.382		0.415
6–12	328	110 (34%)		131 (40%)		104 (32%)	
13–24	658	209 (32%)		282 (43%)		192 (29%)	
Gender			0.339		0.784		0.578
Male	513	173 (34%)		217 (42%)		150 (29%)	
Female	473	146 (31%)		196 (41%)		146 (31%)	
Premature			0.273		0.570		0.247
Yes	48	19 (40%)		22 (46%)		18 (38%)	
No	938	300 (32%)		391 (42%)		278 (30%)	
Ethnicity			0.773		0.918		0.532
Han ethnicity	894	288 (32%)		374 (42%)		271 (30%)	
Ethnic minority	92	31 (34%)		39 (42%)		25 (27%)	
Have siblings			0.492		0.422		0.593
Yes	681	225 (33%)		291 (43%)		208 (31%)	
No	305	94 (31%)		122 (40%)		88 (29%)	
***Household characteristics***							
Primary caregiver			<0.01		<0.01		<0.01
Mother	777	230 (30%)		294 (38%)		217 (28%)	
Others	209	89 (43%)		119 (57%)		79 (38%)	
Caregiver age (years)			<0.01		<0.01		<0.01
<25	305	92 (30%)		125 (41%)		95 (31%)	
25~45	509	146 (29%)		182 (36%)		132 (26%)	
>45	172	81 (47%)		106 (62%)		69 (40%)	
Caregiver educational level			<0.01		<0.01		0.965
under 9 years	574	215 (37%)		264 (46%)		172 (30%)	
9 years and above	412	104 (25%)		149 (36%)		124 (30%)	
Paternal educational level			<0.01		<0.01		0.531
under 9 years	538	205 (38%)		251 (47%)		166 (31%)	
9 years and above	448	114 (25%)		162 (36%)		130 (29%)	
Father at home			0.320		0.540		0.063
Yes	280	84 (30%)		113 (40%)		72 (26%)	
No	706	235 (33%)		300 (42%)		224 (32%)	
Household asset index			<0.01		<0.01		0.660
Bottom 75%	737	263 (36%)		329 (45%)		224 (30%)	
Top 25%	249	56 (22%)		84 (34%)		72 (29%)	

Note. Data source is author’s survey.

**Table 4 ijerph-18-00197-t004:** Analysis of the influence of child and household characteristics on caregiver’s mental health, *n* = 986.

		Depression Score	Anxiety Score	Stress Score
		(1)	(2)	(3)
(1)	Age	0.00	0.04	−0.01
	(in months)	(0.05)	(0.05)	(0.05)
(2)	Male	−0.42	−0.19	−0.52
	(1 = yes)	(0.44)	(0.48)	(0.59)
(3)	Premature	2.21	1.74	2.56
	(1 = yes)	(1.24)	(1.07)	(1.38)
(4)	Han nationality	−1.39	−1.39	−0.13
	(1 = yes)	(1.14)	(1.45)	(1.44)
(5)	Have siblings	0.34	−0.01	0.30
	(1 = yes)	(0.52)	(0.50)	(0.65)
(6)	Primary caregiver	−2.27	−2.67 **	−3.39 **
	(1 = mother)	(1.31)	(1.21)	(1.62)
(7)	Caregiver’s age	0.01	0.01	-0.02
	(in years)	(0.06)	(0.05)	(0.07)
(8)	Caregiver’s educational level	0.35	0.41	−1.13
	(1 = under 9 years)	(0.87)	(0.74)	(0.80)
(9)	Paternal educational level	1.81 ***	1.03	0.53
	(1 = under 9 years)	(0.61)	(0.53)	(0.67)
(10)	Father stay at home	−0.18	0.31	−0.65
	(1 = yes)	(0.59)	(0.65)	(0.73)
(11)	Household asset index	−0.96 ***	−0.69 ***	−0.19
		(0.31)	(0.22)	(0.25)
(12)	Adj. *R*^2^	0.07	0.05	0.01

Note. Data source is author’s survey. All standard errors account for clustering at the village level. ** *p* < 0.05; *** *p* < 0.01.

**Table 5 ijerph-18-00197-t005:** Child development outcomes, *n* = 986.

		Frequency (*n*)	Percentage (%)
		(1)	(2)
(1)	Cognitive delay (1 = yes)	513	52
(2)	Language delay (1 = yes)	526	53
(3)	Social-emotional delay (1 = yes)	505	51
(4)	Motor delay (1 = yes)	310	31
(5)	Any of four types of delay (1 = yes)	850	86
(6)	Any of two types of delay (1 = yes)	590	60
(7)	Any of three types of delay (1 = yes)	327	33
(8)	Four types of delay (1 = yes)	87	9

Note. Data source is author’s survey.

**Table 6 ijerph-18-00197-t006:** Correlates between caregiver’s mental health and child development outcomes using ordinary least squares, *n* = 986.

		Cognitive Scores	Language Scores	Social-Emotional Scores	Motor Scores
		(1)	(2)	(3)	(4)
(1)	Depression	−2.59 ***	−2.34 ***	−2.86 ***	−2.91 ***
	(1 = yes)	(0.98)	(0.83)	(0.88)	(1.00)
	Controls	YES	YES	YES	YES
	Tester Fixed Effects	YES	YES	YES	YES
	Adj. *R*^2^	0.15	0.29	0.18	0.30
(2)	Anxiety	−0.81	−1.47	−1.48	−2.13 **
	(1 = yes)	(0.88)	(0.87)	(0.79)	(0.98)
	Controls	YES	YES	YES	YES
	Tester Fixed Effects	YES	YES	YES	YES
	Adj. *R*^2^	0.14	0.28	0.17	0.30
(3)	Stress	−1.56	−2.98 ***	−0.40	−2.61**
	(1 = yes)	(1.01)	(0.88)	(0.89)	(1.10)
	Controls	YES	YES	YES	YES
	Tester Fixed Effects	YES	YES	YES	YES
	Adj. *R*^2^	0.14	0.29	0.17	0.30
(4)	Any of mental health problem	−1.87**	−2.00 **	−2.00 **	−2.91 ***
	(1 = depression or anxiety or stress)	(0.88)	(0.81)	(0.81)	(0.89)
	Controls	YES	YES	YES	YES
	Tester Fixed Effects	YES	YES	YES	YES
	Adj. *R*^2^	0.15	0.29	0.18	0.30

Note. Data source is author’s survey. Controls include child’s age, gender, ethnicity, and premature birth; whether the child has siblings; whether the mother is the primary caregiver; caregiver’s age and educational level; paternal educational level; whether the father lives at home; and household asset index. We also control for Bayley tester fixed effects. All standard errors account for clustering at the village level. ** *p* < 0.05; *** *p* < 0.01.

**Table 7 ijerph-18-00197-t007:** Heterogeneous effects of caregiver’s mental health on ECD across certain variables from socioeconomic status, *n* = 986.

		Cognitive Scores	Language Scores	Social-Emotional Scores	Motor Scores
		(1)	(2)	(3)	(4)
	Panel A: caregiver mental health and educational level				
(1)	Caregiver with mental health problem and low educational level	−3.53 **	−4.42 ***	−4.76 ***	−4.12 **
	(1 = yes)	(1.37)	(1.29)	(1.49)	(1.64)
(2)	Caregiver with mental health problem and high educational level	−1.06	−1.64	−3.61 ***	−1.36
	(1 = yes)	(1.13)	(1.22)	(1.26)	(1.36)
(3)	Caregiver without mental health problem and with low educational level	−1.05	−2.16	−3.97 ***	−0.04
	(1 = yes)	(1.17)	(1.30)	(1.37)	(1.50)
	Controls	Yes	Yes	Yes	Yes
	Tester Fixed Effects	Yes	Yes	Yes	Yes
	Adj. *R*^2^	0.15	0.29	0.18	0.30
(4)	*p* value of test (1) = (2)	0.070	0.018	0.379	0.154
(5)	*p* value of test (1) = (3)	0.039	0.035	0.437	<0.01
	Panel B: caregiver mental health and paternal educational level				
(6)	Caregiver with mental health problem and low paternal educational level	−3.40 ***	−3.44 **	−1.46	−3.38 **
	(1 = yes)	(1.08)	(1.35)	(1.15)	(1.30)
(7)	Caregiver with mental health problem and high paternal educational level	−2.22	−2.21	−1.47	−2.66 **
	(1 = yes)	(1.29)	(1.17)	(1.15)	(1.32)
(8)	Caregiver without mental health problem and with low paternal educational level	−1.83	−1.63	1.01	−0.26
	(1 = yes)	(1.23)	(1.23)	(1.28)	(1.34)
	Controls	Yes	Yes	Yes	Yes
	Tester Fixed Effects	Yes	Yes	Yes	Yes
	Adj. *R*^2^	0.15	0.29	0.18	0.30
(9)	*p* value of test (6) = (7)	0.266	0.323	0.998	0.557
(10)	*p* value of test (6) = (8)	0.162	0.078	0.027	<0.01
	Panel C: caregiver mental health and household asset				
(11)	Caregiver with mental health problem and low household asset	−3.89 ***	−6.04 ***	−3.95 ***	−5.38 ***
	(1 = yes)	(1.28)	(1.25)	(1.09)	(1.51)
(12)	Caregiver with mental health problem and high household asset	−2.02	−4.15 ***	−1.56	−5.19 ***
	(1 = yes)	(1.52)	(1.46)	(1.46)	(1.78)
(13)	Caregiver without mental health problem and with low household asset	−2.08	−4.84 ***	−1.81	−3.28 **
	(1 = yes)	(1.17)	(1.25)	(1.25)	(1.51)
	Controls	Yes	Yes	Yes	Yes
	Tester Fixed Effects	Yes	Yes	Yes	Yes
	Adj. *R*^2^	0.14	0.29	0.18	0.30
(14)	*p* value of test (11) = (12)	0.189	0.160	0.070	0.891
(15)	*p* value of test (11) = (13)	0.072	0.163	0.034	0.031

Note. Data source is author’s survey. Controls include child’s age, gender, ethnicity, and premature birth; whether the child has siblings; whether the mother is the primary caregiver; caregiver’s age and educational level; paternal educational level; whether the father lives at home; and household asset index. Low household assets indicates the last 75% families; high household assets indicates the remaining families. We also control for Bayley tester fixed effects. All standard errors account for clustering at the village level. ** *p* < 0.05; *** *p* < 0.01.

## Data Availability

The data presented in this study are available on request from the corresponding author. The data are not publicly available dur to privacy.

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
