# Peer review of "Mental Health Issues among Caregivers of Young Children in Rural China: Prevalence, Risk Factors, and Links to Child Developmental Outcomes"

_ijerph, 2020, doi:10.3390/ijerph18010197_

Round 1
Reviewer 1 Report
Abstract:
Please define ECD in the abstract.
Introduction:
Lines 37-39 need restructuring.
Lines 39-43. Please restructure. I don't understand what is high levels and low levels of ECD.
Line 45: (but not always) is understood.
Line 45-47: Use often/usually in this sentence too.
Lines 93-95: are you sure?
Materials and Methods:
Were the rating scales in English, translated into the regional language on paper, or interpreted by the raters?
Discussion:
Line 415: Please mention the prevalence found in this study
Reviewer 2 Report
This study investigated the prevalence of mental health problems in caregivers of young children in rural China and its correlations with child developmental outcomes. The study findings could increase the awareness on the importance of mental health issues in rural China.
However, the study has serious flaw in the statistical analyses. All the outcomes were presented as mean and SD, which is not applicable to those categorical data. For example, gender should be presented as percentage rather than mean score. Authors presented in the manuscript that 52% of the young children were male, but in the table, it is stated the mean score is 0.52. This is very confusing and incorrect. same to other categorical data. in addition, the mean score of depression, anxiety and stress also having the same problems. if the authors using the total score, then the outcomes shouldn't be only between 0-1. Thus, the analyses and the results presented in the paper is not incorrect and require re-analysis. The authors should understand the basic concept of statistical analysis.
The presentation of the study also required revision. For example, it is not necessary to describe your outline of the paper in a research article (page 3, line 105-107).
Reviewer 3 Report
A generally good paper - For consideration, I recommend the inclusion of research which presents / discusses the conflicting and divergent perspectives in the literature, and to further updated the Abstract / Methodology to note the limitations to your paper as opposed to the back end of the paper.
See the attachment.
